# Scene, Class, Signal: Tri-Level Adaptation for Synthetic-to-Real LiDAR Segmentation

## Abstract

Synthetic LiDAR datasets offer a scalable alternative to costly real-world annotations, but still exhibit a significant domain gap when applied to real-world data. Previous unsupervised domain adaptation (UDA) methods mainly rely on general adaptation strategies, without directly addressing the LiDAR-specific factors causing this gap. In this work, we analyze the synthetic-to-real domain gap from a root-cause-driven perspective. We decompose the components of this gap into three distinct granularities: **scene-level**, **class-level**, and **signal-level**. At the **scene-level**, we address the point structure distortions caused by real-world sensor effects, such as motion blur and rolling shutter. At the **class-level**, we consider that the domain gap varies depending on the structural complexity and dynamicity of each object class. Finally, at the **signal-level**, we tackle the lack of direct, realistic semantic information that corresponds to the synthetic input. To address these challenges, we propose a multi-level adaptation framework. Motivated by our finding that encoder feature statistics (mean and variance) capture point structural domain gap, we propose to employ a *style embedding*. Built from the feature mean and variance, this embedding serves as a domain cue for adversarial learning at the **scene-level** and extends to the **class-level** for category-dependent shifts. At the **signal-level**, we complement this with an *intensity-guided self-training* scheme for handling non-structural gaps, leveraging real LiDAR intensity as weak supervision for synthetic inputs. On *SynLiDAR→SemanticKITTI*, our method achieves 44.7 mIoU, and on *SynLiDAR→SemanticPOSS*, it reaches 51.2 mIoU, setting a new state of the art on both benchmarks. Extensive ablation studies validate each component, confirming our style embedding captures the structural domain gap while our self-training scheme significantly improves adaptation.

## 1 Introduction

LiDAR semantic segmentation (LSS), which aims to assign a semantic label to each 3D point captured by LiDAR, is a fundamental task for scene understanding in autonomous driving. By providing dense and geometrically accurate representations of the environment, LSS plays a key role in enabling high-level decision-making processes such as path planning and obstacle avoidance. Despite its importance, the development of high-performing LSS models remains challenging due to the significant annotation cost associated with acquiring large-scale, labeled real-world data.

Unlike other data domain, collecting LiDAR data for outdoor driving scenes requires extensive physical driving across diverse environments while recording high-frequency 3D point clouds. This process is inherently time-consuming and must be conducted under real traffic conditions, where safety risks are persistently present due to the unpredictable behavior of surrounding agents. Furthermore, the annotation of LiDAR point clouds is labor-intensive, as it often involves manual cross-referencing with RGB images to determine point-wise semantic labels (Behley et al., 2019).

To alleviate the need for physical data acquisition and human annotation, synthetic LiDAR datasets have gained popularity. These datasets enable scalable supervision through simulation, allowing models to be trained on fully annotated synthetic data and then adapted to real domains. However, bridging the synthetic-to-real domain gap is uniquely difficult for LiDAR, as the discrepancies are not limited to texture or appearance (as in images). In particular, the domain shift in LiDAR stems from physical signal formation and sensor characteristics. This makes LiDAR adaptation a different and more complex challenge compared to other modalities.

Specifically, real-world LiDAR data is affected by phenomena such as beam attenuation, scattering, and occlusion (Manivasagam et al., 2023; Hahner et al., 2021; 2022), leading to sparse, noisy, and

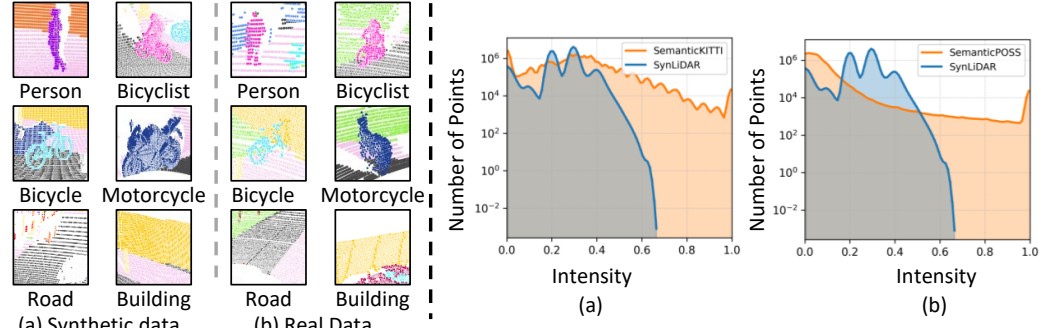

Figure 1: **(Left)** Point structure difference between (a) SynLiDAR and (b) SemanticKITTI. Objects in the first and second row are inherently more dynamic and structurally complex than objects in the third row. As a result, they experience greater point structure distortion, leading to class-dependent variations in the domain gap. **(Right)** Intensity distribution on (a) SynLiDAR & SemanticKITTI and (b) SynLiDAR & SemanticPOSS. A significant difference in intensity distribution can be observed between synthetic and real data.

irregular point distributions. In contrast, synthetic data, typically generated via uniform ray casting, tends to exhibit overly dense and artifact-free geometry. Moreover, real LiDAR sensors introduce temporal distortions such as rolling shutter effects and motion blur (Manivasagam et al., 2020; 2023; Hess et al., 2025), which are rarely modeled in simulation. Intensity distributions also diverge: real beams decay over distance and vary by surface reflectance (Vacek et al., 2021), whereas synthetic intensities are often simplified for efficiency. These observations indicate that the LiDAR domain gap has deeper structural and semantic roots than previously considered.

While recent efforts have proposed various unsupervised domain adaptation (UDA) techniques to bridge this gap, most existing approaches fall short in fully addressing the underlying challenges. Several methods augment inputs with noise (Li et al., 2023), others utilize entropy-based adversarial learning (Yuan et al., 2024), others regress intensity from the target data to create an additional learning signal (Yuan et al., 2023), and a few adopt teacher-student frameworks with self-training. However, these approaches share three key limitations. **(L1)** They rely on indirect domain cues, such as noise and entropy, that reflect consequences of the domain gap rather than its root cause. **(L2)** They apply domain adaptation uniformly across the entire scene, overlooking the fact that domain shifts are often class-dependent. As shown on the left of Fig.1, the severity of the gap varies across object categories, making global adaptation insufficient. **(L3)** Although the intensity value of the target data implicitly contains the semantics of real data, this information is not utilized for learning from synthetic data inputs.

In this paper, we take a root-cause-driven perspective and propose a *coarse-to-fine LiDAR adaptation framework* that systematically addresses the domain gap along three levels of granularity: **scene-level**, **class-level**, and **signal-level**.

In **scene-level**, inspired by prior findings that feature statistics in encoder layers capture the "style" of the input (Huang & Belongie, 2017) and our findings, we hypothesize that such feature statistics of a LiDAR segmentation model can effectively represent point structure differences–such as variations in point density, motion blur, and rolling shutter artifacts–as a form of style.

In **class-level**, following prior work (Manivasagam et al., 2023), we observed that variations in object motion and geometric structure across semantic categories lead to class-dependent domain shifts (the left of Fig. 1). To address this, we extend the use of feature statistics from a scene-level representation to a class-wise formulation. This design allows the model to more precisely capture class-specific structural differences, thereby improving its ability to mitigate class-dependent domain shifts. Moreover, semantic similarity among classes naturally implies similarity in their complexity and dynamics. Thus, to effectively address class-dependent domain gaps, style representations should be aligned among visually similar classes. To this end, we adopt a class-hierarchical formulation that aligns styles within superclasses (e.g., `bicycle` and `motorcycle`), thereby capturing structural similarities among related classes. This formulation ensures more consistent adaptation and improves model generalization. Note that in row 1 and row 2 on the left of Fig. 1, we visualize examples belonging to the same superclass to illustrate this concept.

In **signal-level**, overfitting to synthetic data arises primarily from the absence of supervision in the real domain in UDA setting. To compensate for the lack of real labels, prior works (Yuan et al., 2023; Viswanath et al., 2025) leverage intensity measurements from real LiDAR scans as an auxiliary learning signal. However, given the fact that intensity contains implicit semantic information (Song et al., 2002; Naich & Carrión, 2024; Viswanath et al., 2025), utilizing this information for synthetic data supervision could be more effective in reducing the domain gap. We advance prior work in this direction. We construct a self-training mechanism that encourages the synthetic data input to follow the real LiDAR intensity distribution, which mitigates the synthetic-to-real gap.

We validate these hypotheses through extensive experiments. Our method achieves state-of-the-art performance, reaching 44.7 mIoU on the SynLiDAR-to-SemanticKITTI benchmark and 51.2 mIoU on the SynLiDAR-to-SemanticPOSS benchmark. Ablation studies further confirm the effectiveness of each component, and the method shows robustness to hyperparameter choices.

In summary:

- We define the domain gap using feature statistics extracted from the encoder of the segmentation model.
- We introduce class-wise style embeddings to account for class-dependent point structure distortions.
- Through our intensity-guided self-training strategy, we encouraged the synthetic data input to follow the real intensity distribution.
- Our one-stage framework achieves state-of-the-art performance on two major synthetic-to-real benchmarks.

## 2 RELATED WORKS

### 2.1 LiDAR SEMANTIC SEGMENTATION

LiDAR semantic segmentation (LSS) methods can be categorized into three groups: point-based, projection-based, and voxel-based approaches.

**Point-based methods** Qi et al. (2017); Thomas et al. (2019); Zhao et al. (2021); Thomas et al. (2024); Choe et al. (2022) directly process LiDAR points. Despite preserving geometric details, these methods are computationally intensive and slow at inference, limiting suitability for dynamic scenes.

**Projection-based methods** Milioto et al. (2019); Ando et al. (2023); Kong et al. (2023a); Xu et al. (2025) convert 3D LiDAR points into 2D representations, typically via spherical projection, allowing efficient use of 2D CNNs. Milioto et al. (2019) pioneered using spherical-projected range images for semantic segmentation. Ando et al. (2023) improved performance by fine-tuning a ViT pretrained on images. Kong et al. (2023a) enhanced projection-based methods with specialized data augmentation. Xu et al. (2025) edited spherical projection with an MLP-based approach, preserving all input point information.

**Voxel-based methods** Choy et al. (2019); Zhou et al. (2020); Lai et al. (2023) voxelize 3D points for computational efficiency and parallel processing. Choy et al. (2019) introduced sparse 3D convolution, and Tang et al. (2020) integrated point-wise features to reduce information loss. Zhou et al. (2020) proposed cylindrical voxelization to address varying point densities. Lai et al. (2023) employed radial window partitioning to capture long-range context. Voxel-based methods balance segmentation quality and inference speed effectively, making them popular for LiDAR segmentation.

### 2.2 SYNTHETIC LiDAR SIMULATIONS

To address the difficulty of collecting large-scale driving scene LiDAR data in the real world, existing studies have attempted to simulate realistic LiDAR data (Manivasagam et al., 2020; 2023; Xiao et al., 2022; Vacek et al., 2021). Manivasagam et al. (2020) proposed a method for generating realistic synthetic LiDAR data by pairing each sample with corresponding real LiDAR data. Manivasagam et al. (2023) generated realistic outdoor LiDAR data by transforming spherical-projected range images to match real data, incorporating effects such as motion blur and rolling shutter. Xiao et al. (2022) used Unreal Engine to synthesize LiDAR data and applied a generative model to simulate point drops, producing more realistic patterns. Vacek et al. (2021) focused on the intensity

gap between synthetic and real data, and used a neural network to generate realistic intensity distributions for ray-casted LiDAR points. Despite these efforts, simulating realistic LiDAR remains computationally expensive and requires modeling all relevant domain gap components. Therefore, synthetic-to-real unsupervised domain adaptation is essential for practical deployment.

### 2.3 SYNTHETIC-TO-REAL UNSUPERVISED DOMAIN ADAPTATION FOR LiDAR SEMANTIC SEGMENTATION

Previous works have addressed domain adaptation by adding noise to the input data (Li et al., 2023), interpreting entropy as a cue for domain gap and minimizing it through adversarial learning, or enhancing teacher-student architectures via self-training mechanisms (Yuan et al., 2024). Additionally, other studies have sought to learn implicit semantic information by training on the intensity values of the target dataset (Yuan et al., 2023). Rather than relying on noise or entropy as the primary source of the domain gap, we extract style embeddings from the segmentation model to directly characterize structural discrepency and reduce the domain gap. Furthermore, unlike prior approaches, we incorporate point intensity within a self-training strategy to be aware of implicit semantic information of the real domain from synthetic input and to prevent overfitting to the synthetic domain.

## 3 MOTIVATION

As discussed in Sec. 1, we take a root-cause-driven perspective and propose a novel LiDAR adaptation framework that systematically addresses the domain gap along three levels of granularity: **scene-level**, **class-level**, and **signal-level**. We must address three key challenges to achieve this: **(scene-level)** discovering domain cues that explicitly capture point structure differences, the main cause of the synthetic-to-real domain gap. **(class-level)** modeling these domain cues from the scene level to the class level to capture class-dependent domain shifts. **(signal-level)** finding a way to utilize intensity, which contains implicit semantic information, for synthetic data input.

**Scene-level Domain Cues from "Style".** Input data for UDA task in LSS is typically limited to `xyz` coordinates. This provides no appearance information. We decompose LiDAR input into two conceptual components: "content" and "style". Content refers to the object's identity, whereas style defines its structural representation, like point density, motion blur, and rolling shutter distortion. Inspired by Huang & Belongie (2017), we find that feature statistics effectively capture this style information through toy experiments. We compute the mean and standard deviation at various feature levels to form a compact "style embedding". This embedding serves as a transferable domain cue, reflecting the synthetic-to-real domain gap. We conducted toy experiments to validate the rationale, and details are provided in the supplementary material.

**Scene-level Domain Cues to Class-level Domain Cues.** As shown in the left Fig. 1, classes such as `person`, `bicyclist`, `bicycle`, and `motorcycle` exhibit greater point structure distortion than classes like `road` or `building`. This is because these classes are typically dynamic and geometrically complex, making them more susceptible to motion blur and rolling shutter artifacts compared to static objects, as mentioned in Manivasagam et al. (2023). This motivates the use of class-aware domain cues to better capture class-specific structural differences. Additionally, semantically similar classes tend to share similar levels of dynamicness and complexity. For instance, classes like `car` and `truck` share visual and structural properties, thus also share similar dynamics and complexity. Consequently, features of synthetic `car` points should align not only with real `car` points but also with real `truck` points, facilitating more effective domain adaptation. Motivated by this observation, we introduce class-hierarchical domain cues that leverage semantic similarity among classes, enabling more precise modeling of class-dependent point structure discrepancies beyond global scene-level domain cues. We conducted toy experiments to validate the rationale behind these two domain cues. Further details are provided in the supplementary material.

**Utilizing LiDAR Intensity for Synthetic Inputs.** In conventional UDA settings, label supervision is applied only to the synthetic domain. This imbalance in learning signal often causes the segmentation model to overfit to synthetic data, and addressing this issue is non-trivial due to the lack of annotations in real data. However, LiDAR sensors naturally capture point-wise intensity values, which can serve as auxiliary information without requiring manual annotation. Since intensity values inherently reflect class-specific characteristics through their dependence on distance and object reflectivity, they implicitly encode semantic class information (Song et al., 2002; Naich & Carrión, 2024; Viswanath et al., 2025). Accordingly, treating intensity from real scans as labels allows the

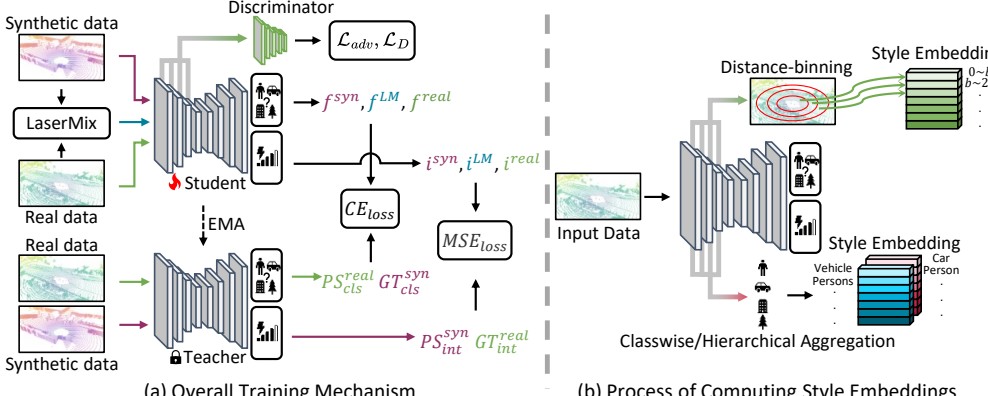

(a) Overall Training Mechanism   (b) Process of Computing Style Embeddings

Figure 2: (a) Overview of our method. (b) Process of computing style embeddings. Our method extracts scene-wise, class-wise style embeddings $(SSE, CSE)$ for adversarial learning. In parallel, we add an intensity prediction head to the feature extractor, enabling intensity self-training by predicting LiDAR point intensities. This head produces outputs $i^{syn}$, $i^{LM}$, and $i^{real}$, corresponding to synthetic, LaserMix-ed, and real inputs, respectively. $f^{syn}$, $f^{LM}$, and $f^{real}$ are the original outputs obtained from the classifier head. $PS_{cls}^{real}$ denotes the pseudo-labels obtained from the teacher model for real data, and $PS_{int}^{syn}$ represents the pseudo-intensities generated by the teacher model for synthetic data. $GT_{cls}^{syn}$ is the ground-truth class label from the synthetic dataset, and $GT_{int}^{real}$ is the ground-truth intensity from real data.

model to learn implicit semantic information. Thus, prior works have utilized intensity to generate additional learning signals (Yuan et al., 2023; Viswanath et al., 2025). However, these studies did not attempt to adapt synthetic inputs using an intensity-based learning signal. This is because synthetic and real data have noticeably different intensity distributions, as shown in the right of Fig. 1. Consequently, using raw synthetic intensity of synthetic data as a learning signal can interfere with learning semantic cues from real data. To address this limitation, we extend prior work by proposing a self-training mechanism. This mechanism generates real-like pseudo-intensity for synthetic inputs and utilizes it during training. The strategy enables the model to learn semantic cues from real data and prevents overfitting to synthetic data.

## 4 METHODS

### 4.1 OVERVIEW

Fig. 2 shows the overview of the proposed framework. We build upon SAC-LM (Yuan et al., 2024), a recent self-training-based domain adaptation framework for LSS. SAC-LM, an extension of Laser-Mix (Kong et al., 2023b), is a method for 3D segmentation that reduces the domain gap. It achieves this by mixing target scans with synthetic source-like scans generated through point drop. A KL-based loss ensures consistency between predictions on raw and source-like target inputs. For a comprehensive methodology, please refer to Yuan et al. (2024). To enable intensity-guided adaptation, we augment the baseline with an additional MLP head dedicated to intensity prediction. This head produces outputs $i^{syn}$, $i^{LM}$, and $i^{real}$, corresponding to synthetic, LaserMix-ed, and real inputs, respectively. In parallel, $f^{syn}$, $f^{LM}$, and $f^{real}$ are the original outputs obtained from the classifier head.

During self-training, we extract two types of style embeddings, scene-wise and class-wise from synthetic, real, and LaserMix-ed data. These embeddings are then used in an adversarial learning framework to align domain-specific styles across domains. The teacher model is updated as the exponential moving average (EMA) of the student model

### 4.2 ADVERSARIAL LEARNING WITH STYLE EMBEDDINGS

#### 4.2.1 SCENE-WISE STYLE EMBEDDING (SSE).

This embedding aims to capture domain gap factors that are class-agnostic and scene-wise, such as motion blur and rolling shutter distortions. As shown in Fig. 2(b), we divide the 3D points into radial distance bins, using a maximum distance threshold $d$ and a binning interval $b$. Since point sparsity significantly varies with depth, we explicitly consider this variability by extracting style embeddings

separately for each distance bin. For each encoder block, we track the downsampled point features and retrieve their corresponding $xyz$ coordinates. Within each distance bin $b_i$, we compute the mean and standard deviation of point features and concatenate them across all encoder blocks to form the final style embeddings $SSE_{\text{mean}} \in \mathbb{R}^{d/b,D}$ and $SSE_{\text{std}} \in \mathbb{R}^{d/b,D}$. These embeddings are then passed to a domain discriminator to facilitate adversarial learning. The formulation is expressed as follows:

$$SSE_{\text{mean}}^i = \bigoplus_{k=1}^{B} \mathbb{E}_p[f_k(x)], \quad SSE_{\text{std}}^i = \bigoplus_{k=1}^{B} \text{std}_p[f_k(x)]. \tag{1}$$

$B$ is the number of encoder blocks. $\bigoplus$ denotes the concatenation of the mean and standard deviation from each encoder block. $i \in \{\text{real}, \text{LM}, \text{syn}\}$ indicates data domains. $D$ represents the total dimension after concatenation. $f_k(\cdot)$ refers to the $k$-th encoder block, and $p$ represents an individual point.

### 4.2.2 CLASSWISE STYLE EMBEDDING (CSE).

As discussed in Sec. 3, point structure distortions differ across semantic classes. To capture such class-specific variations, we extend the scene-wise style embedding by also incorporating label information. For each encoder block, we downsample the labels consistently with the downsampling of point features, thereby tracking labels corresponding to each point feature. Then the mean and standard deviation of point features are computed within each class. Finally, we concatenate these statistics across all classes to obtain the classwise style embeddings $CSE_{\text{mean}}^{cls} \in \mathbb{R}^{C,D}$ and $CSE_{\text{std}}^{cls} \in \mathbb{R}^{C,D}$. Fig. 2 (b) visualizes this process. The formulation is expressed as follows:

$$CSE_{\text{mean}}^{cls,i} = \bigoplus_{k=1}^{B} \mathbb{E}_{p \in c}[f_k(x)], \quad CSE_{\text{std}}^{cls,i} = \bigoplus_{k=1}^{B} \text{std}_{p \in c}[f_k(x)]. \tag{2}$$

$C$ is number of classes and $p \in c$ indicates that point $p$ belongs to a specific class $c$. In the real domain, encoder features are grouped by class using the pseudo-labels.

As mentioned in Sec. 3, semantically similar classes share dynamics and structural complexity. By leveraging this fact, we assume that classes within the same superclass exhibit similar domain gaps, which enables more effective compensation of class-dependent point structure differences during domain adaptation. In this reason, we further introduce class-hierarchical style embeddings to extend class-dependent feature alignment at the superclass level. To implement this, we define a class hierarchy automatically through GPT-4o. We filtered contradictory findings based on previous research that utilized class hierarchies (Kim et al., 2023; Park et al., 2025). The resulting hierarchy is presented in Table 1. Then, we aggregate features across superclass groups using the same mean and standard deviation computation used for $CSE$. This results in class-hierarchical style embeddings $CSE_{\text{mean}}^{hier} \in \mathbb{R}^{H,D}$ and $CSE_{\text{std}}^{hier} \in \mathbb{R}^{H,D}$. The formulation is expressed as follows:

$$CSE_{\text{mean}}^{hier,i} = \bigoplus_{k=1}^{B} \mathbb{E}_{p \in h}[f_k(x)], \quad CSE_{\text{std}}^{hier,i} = \bigoplus_{k=1}^{B} \text{std}_{p \in h}[f_k(x)]. \tag{3}$$

$H$ is the number of superclasses, defined in class hierarchy. $p \in h$ indicates that point $p$ belongs to a specific superclass $h$. Unlike Park et al. (2025), our method does not directly utilize the class as supervision. Instead, our approach extends the scene-wise domain gap to a class-wise domain gap.

### 4.2.3 STYLE EMBEDDING-BASED ADVERSARIAL LEARNING.

We perform adversarial learning using the previously described scene-wise and classwise, which are extracted from each synthetic, real, and LaserMix-ed data pipeline. By performing adversarial learning on these style embeddings, we obtain segmentation features that are invariant to domain gaps caused by differences in point structure. The formulation is expressed as follows:

$$\mathcal{L}_{\text{adv}} = -\frac{1}{N^{\text{real}}} \sum_i \|D(SE_i^{\text{real}})\|^2 - \frac{1}{N^{\text{LM}}} \sum_i \|D(SE_i^{\text{LM}})\|^2, \tag{4}$$

$$\mathcal{L}_{\text{D}} = -\frac{1}{N^{\text{syn}}} \sum_i \|D(SE_i^{\text{syn}})\|^2 - \frac{1}{N^{\text{real}}} \sum_i \|D(SE_i^{\text{real}}) - 1\|^2 - \frac{1}{N^{\text{LM}}} \sum_i \|D(SE_i^{\text{LM}}) - 1\|^2. \tag{5}$$

$SE \in \{SSE, CSE\}$ denotes the chosen style-embedding type, and $i \in \{mean, std\}$ selects either the mean or standard-deviation branch. For the discriminator architecture and adversarial loss, we adopt the same settings as in Yuan et al. (2024).

| Superclass | SemanticKITTI Classes |
|---|---|
| Vehicle | car, bicycle, motorcycle, truck, other-vehicle |
| Person | pedestrian, bicyclist, motorcyclist |
| Traffic Element | pole, traffic-sign |
| Pavement | road, parking, sidewalk, other-ground |
| Natural | vegetation, trunk, terrain |
| Structure | building, fence |

| Superclass | SemanticPOSS Classes |
|---|---|
| Vehicle | car, bike |
| Person | rider, pedestrian |
| Traffic Element | traffic-sign, pole |
| Object | trash-can, cone-stone |
| Pavement | ground |
| Natural | trunk, plant |
| Structure | building, fence |

Table 1: Superclass hierarchy for the SynLiDAR-to-SemanticKITTI (top) and SynLiDAR-to-SemanticPOSS (bottom) setups.

### 4.3 INTENSITY-GUIDED SELF-TRAINING

To mitigate overfitting to synthetic data and to enable the model to learn implicit semantic information of real data from synthetic inputs, we propose a self-training mechanism based on LiDAR intensity. An intensity prediction head is appended to the student and teacher models, respectively. Both networks adopt the same segmentation architecture, with the teacher model updated as the exponential moving average (EMA) of the student model. During training, the student model learns to predict real intensity values using ground-truth labels from real LiDAR data. For synthetic data, the teacher model predicts pseudo-intensity values, which the student model then learns to match. The overall training pipeline is illustrated in Fig. 2(a). The formulation is expressed as follows:

$$\mathcal{L}_{IS} = \sum_{k \in \{\text{real,LM,Syn}\}} \left\| \text{IH}(f(x)) - i^k \right\|^2. \tag{6}$$

IH denotes the intensity head, and $f$ denotes the final output of the feature extractor. $i$ denotes the intensity corresponding to the ground-truth intensity of real data, the pseudo-intensity of synthetic data, and the intensity values of LaserMix-ed points.

### 4.4 OPTIMIZATION AND INFERENCE

The total loss for student model combines classification, intensity, and adversarial terms mentioned above:

$$\mathcal{L} = \mathcal{L}_{cls} + \lambda_{int}\mathcal{L}_{IS} + \lambda_{adv}\mathcal{L}_{\text{adv}}. \tag{7}$$

$\lambda_{int}$ and $\lambda_{adv}$ were 10 and 1e-4 for main experiments, respectively. The teacher updates by EMA of the student. At test time, we use the student alone without an intensity head.

## 5 EXPERIMENTS

### 5.1 EXPERIMENTS SETTINGS

We use the SynLiDAR dataset (Xiao et al., 2022) as our synthetic source domain. SynLiDAR is a large-scale synthetic point cloud dataset generated from diverse urban driving scenes built on the Unreal Engine 4 platform. The LiDAR sensor is simulated based on the Velodyne HDL-64E, producing high-resolution point clouds with up to 64 beams. For real target domains, we use the SemanticKITTI (Behley et al., 2019) and SemanticPOSS (Pan et al., 2020) datasets. SemanticKITTI is collected using the Velodyne HDL-64E sensor with 60 beams and contains real-world point cloud data collected across 22 driving sequences in urban environments. SemanticPOSS uses the Velodyne Alpha Prime sensor with 40 beams and contains point clouds captured within a university campus. Following the adaptation setup in previous work (Yuan et al., 2024), we downsample the SynLiDAR point clouds to 40 beams when training on the SemanticPOSS target. Consistent with Yuan et al. (2024), we adopt MinkUnet32 (Choy et al., 2019) as base segmentation architecture. We train our models for 200,000 iterations, consistent with DGT-ST. Each experiment is conducted on a single NVIDIA A6000 GPU.

| Methods | Mech. | car | bi.cle | mt.cle | truck | oth-v. | pers. | bi.clst | mt.clst | road | parki. | sidew. | other-g. | build. | fence | veget. | trunk | terr. | pole | traf. | mIoU |
|---|---|---|---|---|---|---|---|---|---|---|---|---|---|---|---|---|---|---|---|---|---|
| Source only | - | 35.9 | 7.5 | 10.7 | 0.6 | 2.9 | 13.3 | 44.7 | 3.4 | 21.8 | 6.9 | 29.6 | 0.0 | 34.1 | 7.4 | 62.9 | 26.0 | 35.5 | 30.3 | 14.1 | 20.4 |
| AdaptSegNet Tsai et al. (2018) | A | 52.1 | 10.8 | 11.2 | 2.6 | 9.6 | 15.1 | 35.9 | 2.6 | 62.2 | 10.4 | 41.3 | **0.1** | 58.1 | 17.1 | 68.0 | 38.4 | 38.7 | 35.9 | 20.4 | 27.9 |
| CLAN Luo et al. (2019) | A | 51.0 | 15.8 | 16.8 | 2.2 | 7.8 | 18.7 | 46.8 | 3.0 | 68.9 | 11.1 | 44.9 | 0.1 | 59.6 | 17.5 | 71.7 | 41.1 | 44.0 | 37.7 | 19.8 | 30.5 |
| ADVENT Vu et al. (2019) | A | 59.9 | 13.8 | 14.6 | 3.0 | 8.0 | 17.7 | 45.8 | 3.0 | 67.6 | 11.3 | 45.6 | 0.1 | 61.7 | 15.8 | 72.4 | 41.5 | 47.0 | 34.5 | 15.3 | 30.5 |
| FADA Wang et al. (2020) | A | 49.9 | 6.7 | 5.1 | 2.5 | 10.0 | 5.7 | 26.6 | 2.3 | 65.8 | 10.8 | 37.8 | 0.1 | 60.3 | **21.5** | 60.4 | 37.2 | 31.9 | 35.4 | 17.4 | 25.6 |
| MRNet Zheng & Yang (2019) | A | 49.5 | 11.0 | 12.2 | 2.2 | 8.6 | 16.0 | 46.4 | 2.7 | 60.0 | 10.5 | 41.9 | 0.1 | 55.1 | 16.5 | 68.1 | 38.0 | 40.7 | 36.5 | 20.8 | 28.3 |
| PMAN Yuan et al. (2023) | A | 71.0 | 14.9 | 24.8 | 1.6 | 6.6 | 23.6 | 61.1 | **5.5** | 75.3 | 10.5 | 54.1 | 0.1 | 47.9 | 17.4 | 69.6 | 38.6 | **61.5** | 37.0 | 18.6 | 33.7 |
| PCAN Yuan et al. (2024) | A | 85.0 | **17.5** | 27.4 | 10.4 | **11.9** | 27.5 | 63.7 | 2.6 | 78.1 | 13.5 | 50.1 | 0.1 | 68.5 | 20.0 | 76.2 | 41.3 | 45.7 | 41.0 | 21.8 | 37.0 |
| CoSMix Saltori et al. (2022) | S | 56.4 | 10.2 | 20.8 | 2.1 | **13.0** | 25.6 | 41.3 | 2.2 | 67.4 | 8.2 | 43.4 | 0.0 | 57.9 | 12.2 | 68.4 | 44.8 | 35.0 | 42.1 | 17.0 | 29.9 |
| LaserMix Kong et al. (2023b) | S | 90.3 | 7.8 | 37.2 | 2.3 | 2.4 | 40.6 | 49.1 | 5.1 | 80.5 | 9.9 | 57.4 | 0.0 | 57.6 | 3.4 | 77.6 | 46.6 | 60.1 | 42.0 | 13.6 | 36.0 |
| DGT-ST Yuan et al. (2024) | S | **92.9** | 17.3 | 43.4 | **15.0** | 6.1 | 49.2 | 54.2 | 4.2 | 86.4 | 19.1 | 62.3 | 0.0 | **78.2** | 9.2 | **83.3** | 56.0 | 59.1 | 51.2 | 32.3 | 43.1 |
| Ours | A+S | 90.6 | 14.4 | 51.9 | 11.3 | 4.3 | 59.9 | 66.7 | 3.2 | **88.8** | 27.4 | 64.7 | 0.0 | 69.9 | 11.7 | 79.5 | 60.1 | 55.0 | 51.3 | 39.2 | **44.7** |

Table 2: Comparison results on *SynLiDAR → SemanticKITTI* benchmark. **Bold**=highest, underline=second highest in each column. A/S denotes adversarial training (A) / self-training (S).

| Methods | Mech. | bi.clst | car | trunk | veget. | traf. | pole | garb. | build. | cone. | fence | bi.cle | ground | pers. | mIoU |
|---|---|---|---|---|---|---|---|---|---|---|---|---|---|---|---|
| Source only | - | 47.2 | 43.6 | 37.8 | 70.3 | 11.1 | 33.8 | 19.5 | 67.9 | 11.2 | 19.9 | 9.6 | 77.9 | 47.8 | 38.3 |
| AdaptSegNet Tsai et al. (2018) | A | 43.9 | 48.2 | 39.0 | 69.6 | 15.5 | 33.6 | 21.3 | 64.3 | 12.7 | 25.0 | 11.6 | 76.0 | 49.9 | 39.3 |
| CLAN Luo et al. (2019) | A | 43.9 | 46.6 | 41.3 | 71.0 | 15.1 | 34.3 | 20.4 | 69.6 | 9.5 | 23.2 | 12.0 | 75.1 | 51.3 | 39.5 |
| ADVENT Vu et al. (2019) | A | 44.6 | 47.6 | 40.3 | 71.2 | 15.6 | 35.6 | 22.0 | 69.6 | 10.6 | 25.9 | 10.4 | 76.7 | 52.3 | 40.1 |
| FADA Wang et al. (2020) | A | 39.6 | 41.2 | 38.8 | 69.2 | 16.3 | 32.1 | 18.1 | 67.9 | 11.5 | 22.0 | 13.0 | 71.4 | 47.9 | 37.6 |
| MRNet Zheng & Yang (2019) | A | 43.5 | 47.2 | 39.1 | 70.4 | 15.5 | 32.8 | 22.0 | 66.1 | 13.2 | 24.2 | 11.2 | 76.8 | 50.0 | 39.4 |
| PMAN Yuan et al. (2023) | A | 52.6 | 61.5 | 44.8 | 75.1 | 18.8 | 36.5 | 21.4 | 74.7 | 18.3 | 25.8 | 37.5 | 73.7 | 61.9 | 46.5 |
| PCAN Yuan et al. (2024) | A | 48.6 | 62.1 | 37.5 | 74.0 | 23.9 | 31.4 | 22.2 | 76.9 | 6.5 | 41.9 | 11.9 | 79.1 | 61.2 | 44.4 |
| CoSMix Saltori et al. (2022) | S | 53.6 | 47.6 | 44.8 | 75.1 | 16.8 | 37.9 | 25.3 | 72.7 | 19.9 | 39.7 | 10.8 | 80.0 | 56.5 | 44.6 |
| LaserMix Kong et al. (2023b) | S | 58.4 | 61.3 | 47.7 | 69.0 | 21.9 | 39.5 | 30.9 | 61.0 | 16.1 | 36.5 | 7.1 | 79.5 | 62.6 | 45.5 |
| DGT-ST Yuan et al. (2024) | S | 55.1 | 70.7 | 46.1 | 74.2 | 30.1 | 36.3 | 44.1 | 81.0 | 4.3 | 62.8 | 10.3 | 78.5 | 67.2 | 50.8 |
| Ours | A+S | 63.5 | 71.2 | 50.4 | 75.3 | 21.7 | 37.0 | 47.1 | 81.4 | 1.0 | 60.6 | 7.5 | 79.0 | 69.7 | 51.2 |

Table 3: Comparison results on *SynLiDAR → SemanticPOSS* benchmark. **Bold**=highest, underline=second highest in each column. A/S denotes adversarial training (A) / self-training (S).

## 5.2 MAIN EXPERIMENTS

**SynLiDAR-to-SemanticKITTI.** As shown in Table 2, our method achieves 44.7 mIoU on the SynLiDAR-to-SemanticKITTI benchmark, outperforming the previous state-of-the-art DGT-ST by +1.6 mIoU. Notably, we observe substantial gains in complex and dynamic object classes such as person (**+10.7**), motorcycle (**+8.5**), and bicyclist (**+12.5**) mIoU. These classes are characterized by irregular motion and geometrically complex shapes, which typically result in larger domain gaps. The consistent improvements on these classes indicate that our method effectively captures point structure differences, a key factor in the domain gap. This also confirms that the classwise style embeddings successfully reduce the additional domain shift caused by class-specific complex and dynamic natures. In addition, we observe gains of +9.7, +3.2, and +1.9 mIoU in classes such as parking, sidewalk, and road, respectively. These results indicate that our method leads to performance improvements across most semantic classes.

**SynLiDAR-to-SemanticPOSS.** As shown in Table 3, our method achieves 51.2 mIoU on the SynLiDAR-to-SemanticPOSS benchmark, surpassing the previous state-of-the-art by +0.4 mIoU. Compared to SemanticKITTI, the SemanticPOSS dataset uses a 40-beam LiDAR sensor, resulting in sparser point clouds during data acquisition. Consequently, reducing the domain gap is harder and yields modest gains, yet absolute scores stay high because the dataset contains fewer samples and classes. Moreover, SemanticPOSS (Pan et al., 2020) contains mostly static and few scenes and only a few classes, leading to a high baseline mIoU and leaving little room for further improvement. Despite this, our method still achieves substantial improvements in complex object classes, including bicyclist (**+8.4**), garbage can (**+3.0**), and person (**+2.5**) mIoU. We also observe gains in trunk (+4.3), ground (+0.5), and vegetation (+1.1), demonstrating consistent performance benefits across various semantic classes.

## 5.3 ABLATION STUDY

As shown in Table 4, we conduct an ablation study by incrementally adding our proposed components on baseline model, SAC-LM (Yuan et al., 2024). Applying scene-wise adversarial learning yields a +5.6 mIoU gain, demonstrating that the scene-wise style embedding effectively captures global domain gaps, such as those caused by noise, rolling shutter or motion blur. Adding classwise adversarial learning provides an additional +1.1 mIoU gain, indicating the importance of modeling class-specific domain gaps in addition to scene-level shifts and considering dynamicity similarity between classes during adaptation is beneficial, even across domains. Finally, applying intensity

| ID | S-wise | C-wise | Int. | mIoU | Increments |
|----|--------|--------|------|------|------------|
| 0  |        |        |      | 37.0 | -          |
| 1  | ✓      |        |      | 42.6 | +5.6       |
| 2  | ✓      | ✓      |      | 43.7 | +1.1       |
| 3  | ✓      | ✓      | ✓    | **44.7** | +1.0   |

Table 4: Ablation study on the effect of each component. The base model, SAC-LM Yuan et al. (2024), gradually adds Scene-wise (S-wise), Class-wise(C-wise) style embedding-based adversarial learning, and Intensity self-training (Int.) losses.

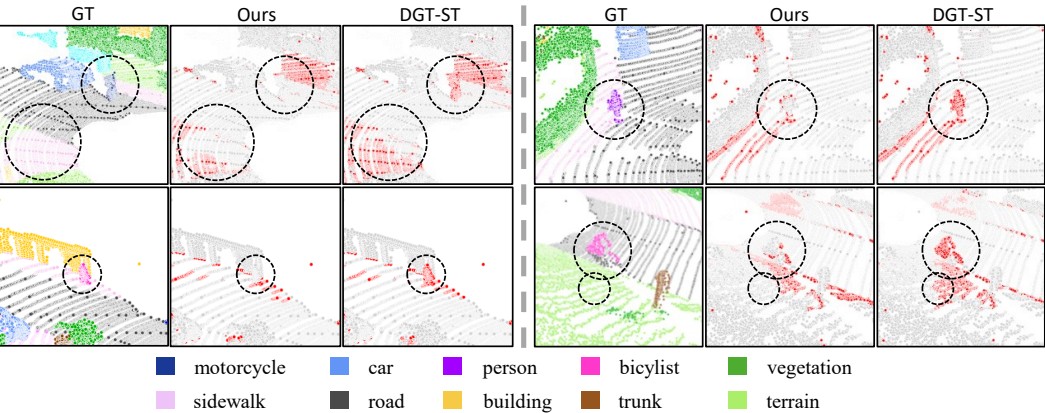

Figure 3: Qualitative results on *SynLiDAR → SemanticKITTI* benchmark. Red=incorrect, Gray=correct. Our method shows improved performance on structurally complex and dynamic objects such as `motorcycle`, `person`, and `bicyclist`.

self-training results in a +1.0 mIoU gain, showing that learning implicit semantic information from synthetic inputs, via pseudo-intensity, is beneficial for domain adaptation.

## 5.4 QUALITATIVE RESULTS

Qualitative results on the *SynLiDAR*-to-*SemanticKITTI* benchmark are presented in Fig. 3. As shown in the left of row 1, our method achieves better segmentation on the `motorcycle` class compared to previous methods, consistent with the analysis in Sec. 5.2. Additionally, improvements on the `person` class can be observed in the left of row 2 and the right of row 1. In the right of row 2, our method also shows better performance on the `bicyclist` class. These results support our findings in Sec. 5.2, where we highlighted that our method yields more accurate predictions for dynamic, and structurally complex objects compared to prior approaches.

## 6 CONCLUSION

We presented a unified framework to address the synthetic-to-real LiDAR domain gap, a key challenge for annotation-efficient semantic segmentation. Unlike general UDA methods, our approach directly targets sensor-induced structural distortions and class-dependent variability, while extending intensity learning to the synthetic data pipeline via self-training. We proposed scene-wise and class-wise style embeddings for adversarial learning and an intensity-guided self-training strategy. Our method achieved state-of-the-art results on two major adaptation benchmarks, highlighting its potential for label-efficient LiDAR segmentation.

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
