# OpenReview forum: "Scene, Class, Signal: Tri‑Level Adaptation for Synthetic‑to‑Real LiDAR Segmentation"
_ICLR.cc/2026/Conference — ICLR 2026 Conference Withdrawn Submission_

### Official Review · Reviewer_GSy6 · 2025-10-27

**Soundness:** 3
**Presentation:** 3
**Contribution:** 3
**Rating:** 6
**Confidence:** 4

**Summary:**

This paper proposes a **Scene–Class–Signal Tri-Level Adaptation framework (TriLA)** for synthetic-to-real LiDAR semantic segmentation.
Unlike prior general UDA approaches that model the domain gap as a whole, TriLA takes a root-cause-driven perspective and decomposes the domain gap into three complementary levels:

(1) **Scene-level:** Constructs style embeddings based on the mean and variance of encoder features to mitigate point-structure distortions caused by motion blur and rolling shutter;

(2) **Class-level:** Models class-dependent structural discrepancies through class-hierarchical alignment;

(3) **Signal-level:** Introduces an *intensity-guided self-training* mechanism that leverages real LiDAR reflectance as weak supervision.

Extensive experiments on SynLiDAR → SemanticKITTI and SynLiDAR → SemanticPOSS demonstrate new state-of-the-art performance.

**Strengths:**

1. The tri-level design (scene, class, signal) is systematic and novel, providing a clear and interpretable framework to simultaneously address global structural differences, class-dependent shifts, and signal-level discrepancies.

2. The formulation of style embeddings (via feature mean and variance) is simple, explainable, and effectively transfers the idea of image style representation to the LiDAR domain.

3. The class-hierarchical alignment automatically constructs class hierarchies, successfully capturing structural similarity among related categories.

4. The experiments are comprehensive, the ablation studies are thorough, and the overall presentation is clear, well-organized, and visually interpretable.

**Weaknesses:**

1.While the hierarchical design is innovative, it lacks visual analysis of feature distributions before and after style embedding (e.g., t-SNE or similarity maps), which would help illustrate the effectiveness and necessity of the proposed design more intuitively. **As a transfer of ideas, this may be important.**

2.Multi-level feature statistics may introduce additional computational overhead; if comparable precedents exist, it would be helpful to include analysis of training time and GPU memory usage.

3.The current experiment only verified the transition scenario from synthesis to reality. If you can further explore the transition from reality to synthesis or across sensors (such as LiDAR with different beam numbers), it will be more convincing.

**Questions:**

The class hierarchy relies on GPT-4o for automatic generation and has certain heuristic features. How does it generalize to different datasets? Is the effect overly dependent on LLM?

---

### Official Review · Reviewer_mRT6 · 2025-10-29

**Soundness:** 2
**Presentation:** 3
**Contribution:** 2
**Rating:** 4
**Confidence:** 4

**Summary:**

This paper addresses the synthetic-to-real domain gap for LiDAR semantic segmentation. The authors posit that this gap is fundamentally rooted in physical sensor properties and signal formation, not just appearance, leading to discrepancies in point structure and signal characteristics. To tackle this, the paper proposes a "Tri-Level Adaptation" framework that decomposes the problem into three granularities: Scene-level, Class-level, Signal-level. The proposed method achieves new state-of-the-art results on the SynLiDAR->SemanticKITTI and SynLiDAR->SemanticPOSS benchmarks.

**Strengths:**

1.	The proposed SSE, CSE and IST seems to be effective. According to the ablation study, each proposed component provides benefit to the baseline and they all bring synergetic gains to the baseline.

2.	The intensity-guided self-training mechanism is novel. By using a teacher model to generate pseudo-intensity targets for synthetic data, the method finds an effective way to bridge the significant intensity distribution gap and learn implicit real-world semantics without real labels.

**Weaknesses:**

1.	The comparison with recent state-of-the-art (SOTA) methods appears insufficient. The most recent baseline used for comparison is DGT-ST (CVPR 2024), which may be outdated. Furthermore, the performance improvement reported in Table 3 (SynLiDAR-to-SemanticPOSS) is a marginal +0.4 mIoU over this baseline. Consequently, the claim to SOTA performance is not fully convincing without comparisons to more recent publications.

2.	The experimental evaluation is limited. The method is only validated on the SemanticKITTI and SemanticPOSS datasets . Given that nuScenes and the Waymo Open Set are also popular and large-scale benchmarks for this task, the reviewer suggests adding experiments on these datasets to better demonstrate the method's generalization capabilities.

3.	The motivation for the proposed Scene-wise Style Embedding (SSE) and Class-wise Style Embedding (CSE) is not sufficiently validated within the main paper. The authors mention that "toy experiments" were conducted to validate the rationale, but these results are relegated to the supplementary material. Including at least a summary of these findings in the main paper would significantly strengthen the motivation for the core components of the method.

4.	The necessity of introducing a superclass hierarchy (for $CSE^{hier}$) is not validated. The reviewer questions whether this component is essential, or if aligning the feature statistics based on individual classes (i.e., standard CSE) would be sufficient. An ablation study on this component is missing.

**Questions:**

1.	Following up on Weakness 1, are there more recent SOTA methods for this task published after DGT-ST (CVPR 2024)? If so, the authors should provide a comparison against these newer works, detailing both the differences in methodology and the quantitative performance.

2.	The caption for Figure 2 is confusingly structured. The sentences from "Our method extracts..." to the end of the caption appear to describe only subfigure (a). The reviewer suggests moving this text to follow the label “(a) Overview of our method.” directly to improve the figure's clarity.

3.	Related to Weakness 4, an ablation study on the effectiveness of the class-hierarchical style embedding ($CSE^{hier}$) is strongly encouraged. This would help validate the necessity and contribution of the superclass concept.

---

### Official Review · Reviewer_f3ng · 2025-10-30

**Soundness:** 3
**Presentation:** 3
**Contribution:** 2
**Rating:** 4
**Confidence:** 3

**Summary:**

This paper tackles unsupervised domain adaptation (UDA) for LiDAR semantic segmentation from synthetic to real data. The authors decompose the domain gap into three granularities and propose a one‑stage framework:

1. Scene‑level: derive scene‑wise style embeddings (SSE) by concatenating the mean and standard deviation of encoder features in radial distance bins; align domains with adversarial learning

2. Class‑level: extend style embeddings to class‑wise and class‑hierarchical forms using pseudo labels and an automatically derived superclass hierarchy。

3. Signal‑level: add an intensity head and perform self‑training so that synthetic inputs learn to follow real‑like intensity distributions via teacher‑generated pseudo‑intensity.

On SynLiDAR→SemanticKITTI the method reports 44.7 mIoU (+1.6 over DGT‑ST), and on SynLiDAR→SemanticPOSS 51.2 mIoU (+0.4), with especially large gains on dynamic classes.

**Strengths:**

1. The paper uses architecture‑agnostic signals. Using encoder feature statistics (mean/std) is lightweight and can be dropped at inference; the intensity head is also training‑only.
2. Clear decomposition with targeted mechanisms. The scene/class/signal split is a crisp way to reason about LiDAR‑specific gaps; each level has a concrete mechanism (SSE/CSE + adversarial; intensity ST). The overall pipeline is easy to follow

**Weaknesses:**

1. Novelty feels incremental relative to prior class‑wise adversarial alignment and “style” statistics.
The use of mean/std as “style” (AdaIN‑style statistics) and adversarial alignment is well‑known; class‑wise/domain alignment echoes CLAN/Classes‑matter. The paper’s novelty is largely in combining these ideas for LiDAR plus an intensity ST twist. That’s a valuable engineering step but less conceptually new than the positioning suggests. More suitable for computer vision focused conferences.

**Questions:**

1. Is there ablation on flat CSE from hierarchical CSE?

---

### Official Review · Reviewer_AMZb · 2025-11-01

**Soundness:** 3
**Presentation:** 3
**Contribution:** 2
**Rating:** 4
**Confidence:** 4

**Summary:**

This paper presents a framework for synthetic-to-real unsupervised domain adaptation (UDA) in LiDAR semantic segmentation.
The authors decompose the LiDAR domain gap into three complementary levels:

> Scene-level: use style embeddings (feature-statistic vectors (mean + variance) extracted from encoder layers) to capture structural distortions caused by sensor effects such as motion blur or rolling shutter.

> Class-level: extend these embeddings to class-wise and class-hierarchical formulations, aligning styles among related categories (e.g., bicycle <--> motorcycle) to mitigate class-dependent shifts.

> Signal-level: introduce an intensity-guided self-training scheme where real-data LiDAR intensities act as weak supervision to regularize synthetic inputs.

The method is implemented atop the SAC-LM/LaserMix UDA framework and tested on SynLiDAR --> SemanticKITTI and SynLiDAR --> SemanticPOSS, reaching 44.7 mIoU and 51.2 mIoU respectively (state-of-the-art results with incremental gains over prior work).

**Strengths:**

+ Well-motivated analysis of LiDAR-specific domain shifts and their physical origins.

+ Simple yet effective integration of adversarial alignment and self-training.

+ Strong empirical execution with ablations and qualitative examples demonstrating improvements on dynamic classes (person, bicycle, motorcycle).

+ Clean, reproducible methodology that could benefit downstream perception systems.

**Weaknesses:**

- Limited representation insight. The paper claims a “root-cause-driven” representation but provides no quantitative or theoretical evidence that the proposed embeddings form more domain-invariant features.

- Narrow scope. Only SynLiDAR --> KITTI/POSS is studied; no cross-sensor or domain-generalization validation.

- Incremental gains. Improvements over prior SOTA are relatively small given the additional complexity.

- Venue fit. The work is framed and evaluated as an applied segmentation system, aligning better with CVPR/IROS than with ICLR’s focus on representation learning.

- Ablation depth. The impact of the GPT-4o-derived hierarchy and hyperparameter sensitivity are not analyzed.

**Questions:**

> Can the authors provide quantitative evidence (e.g., feature-distribution alignment metrics) that the proposed style embeddings genuinely capture structural domain differences?

> How sensitive are results to the radial-binning and class-hierarchy design?

> What is the runtime overhead of extracting and aligning scene-/class-level statistics?

> Could the approach generalize to cross-sensor or cross-weather scenarios beyond SynLiDAR --> KITTI/POSS?

> Would replacing the adversarial discriminator with a contrastive or regularization objective yield similar effects?

---

### Official Review · Reviewer_tzXL · 2025-11-01

**Soundness:** 3
**Presentation:** 3
**Contribution:** 2
**Rating:** 2
**Confidence:** 4

**Summary:**

This paper proposes a multi-level adaptation framework that addresses the synthetic-to-real domain gap in LiDAR segmentation by analyzing and mitigating scene-level, class-level, and signal-level factors, achieving state-of-the-art results on major benchmarks.

**Strengths:**

- The authors’ analysis from the perspectives of scene-level, class-level, and signal-level is reasonable.
- The model achieves state-of-the-art performance.

**Weaknesses:**

1. The authors claim to analyze the LIDAR segmentation task from three perspectives: scene-level, class-level, and signal-level, which is quite interesting. However, I am somewhat disappointed by their proposed solutions, as they mainly utilize techniques e.g., self-training, discriminator, and hierarchical aggregation.  These approaches are essentially no different from those used in natural image segmentation, and do not align with the "root-cause-driven perspective in LIDAR data" emphasized by the authors.

2. The authors compare their method with many early works on LIDAR segmentation (2018-2023). More recent works should be included for comparison and analysis.

3. The ablation study is simple and lacks detailed experimental results and analysis for scene-wise, class-wise, and intensity self-training.

**Questions:**

please refer to the weeknesses

---

### Note · Authors · 2025-11-25

I have read and agree with the venue's withdrawal policy on behalf of myself and my co-authors.